# Gender Disparities after Transcatheter Aortic Valve Replacement with Newer Generation Transcatheter Heart Valves: A Systematic Review and Meta-Analysis

**DOI:** 10.3390/medsci11020033

**Published:** 2023-05-09

**Authors:** Angkawipa Trongtorsak, Sittinun Thangjui, Pabitra Adhikari, Biraj Shrestha, Jakrin Kewcharoen, Leenhapong Navaravong, Somsupha Kanjanauthai, Steve Attanasio, Hammad A. Saudye

**Affiliations:** 1Ascension Saint Francis Hospital, Internal Medicine Residency Program, Evanston, IL 60202, USA; 2Bassett Healthcare Network, Internal Medicine Residency Program, New York, NY 13326, USA; 3Reading Hospital—Tower Health, Internal Medicine Residency Program, West Reading, PA 19611, USA; 4Division of Cardiovascular Medicine, Loma Linda University Health, Loma Linda, CA 92350, USA; 5Division of Cardiovascular Medicine, School of Medicine, University of Utah, Salt Lake City, UT 84112, USA; 6Division of Cardiovascular Medicine, University of Southern California, Los Angeles, CA 90007, USA; 7Rush University Medical Center, Division of Cardiovascular Medicine, Chicago, IL 60612, USA; 8Ascension Saint Francis Hospital, Division of Cardiovascular Medicine, Evanston, IL 60202, USA

**Keywords:** gender disparities, TAVR, newer generation transcatheter heart valves

## Abstract

Previous studies have demonstrated gender disparities in mortality and vascular complications after transcatheter aortic valve replacement (TAVR) with early generation transcatheter heart valves (THVs). It is unclear, however, whether gender-related differences persist with the newer generation THVs. We aim to assess gender disparities after TAVR with newer generation THVs. The MEDLINE and Embase databases were thoroughly searched from inception to April 2023 to identify studies that reported gender-specific outcomes after TAVR with newer generation THVs (Sapien 3, Corevalve Evolut R, and Evolut Pro). The outcomes of interest included 30-day mortality, 1-year mortality, and vascular complications. In total, 5 studies (4 databases) with a total of 47,933 patients (21,073 females and 26,860 males) were included. Ninety-six percent received TAVR via the transfemoral approach. The females had higher 30-day mortality rates (odds ratio (OR) = 1.53, 95% confidence interval (CI) 1.31–1.79, *p*-value (*p*) < 0.001) and vascular complications (OR = 1.43, 95% CI 1.23–1.65, *p* < 0.001). However, one-year mortality was similar between the two groups (OR = 0.78, 95% CI 0.61–1.00, *p* = 0.28). The female gender continues to be associated with higher 30-day mortality rates and vascular complications after TAVR with newer generation transcatheter heart valves, while there was no difference in 1-year mortality between the genders. More data is needed to explore the causes and whether we can improve TAVR outcomes in females.

## 1. Introduction

Previous studies have reported gender disparities in patients with severe aortic stenosis (AS) undergoing transcatheter aortic valve replacement (TAVR) with early generation transcatheter heart valves (THVs) [1,2,3,4,5,6]. The study by Katz et al. reported that women undergoing TAVR experienced higher rates of vascular complications and bleeding after early generation THV including CoreValve, Sapien XT, and Inovare [3]. The authors found that the major vascular complication rates were 6% vs. 11.7%, and the major bleeding rates were 12% vs. 20.6% in men and women, respectively. Another study by Galia et al. similarly found that the female gender was associated with higher early mortality rates compared to male patients after TAVR with early generation THVs [4]. Authors additionally reported that in-hospital and 30-day mortality were 4.3% vs. 8.4%, and 5.4% vs. 9.4% in men and women, respectively. Although the higher risk of outcomes during post-procedure was found to be associated with an increased risk of short-term mortality, it did not translate into a higher risk of outcomes during long-term follow-up [7]. In contrast, several studies found that the female gender appeared to be associated with increased long-term survival compared to men in this population [5,6]. O’Connor et al. reported significantly higher survival rates for women undergoing TAVR with early generation THVs. The 1-year survival estimates were 82% and 78.2% for women and men, respectively.

Since the development of the newer generation THVs such as Edwards Sapien 3, Medtronic Corevalve Evolut R, and Medtronic Evolut Pro, there have been decreased risks of procedural-related complications including vascular complications, bleeding, paravalvular leak (PVL), acute kidney injury, and mortality [8]. However, despite the advancement and better safety profile of the new generation THVs, it is unclear whether differences in outcomes between genders persist with newer generation THVs. As such, we conducted a systematic review and meta-analysis to assess if outcome disparities exist between genders after TAVR with newer generation THVs.

## 2. Methods

### 2.1. Search Strategy

Two investigators (AT and BS) independently performed a thorough systematic search of databases, including the MEDLINE and Embase databases, from inception to April 2023 using a search strategy including the terms “aortic valve replacement”, “transcatheter aortic valve replacement”, and “gender” or “sex”. Full search terms and strategies of databases are described in Appendix A. Additionally, the two investigators hand-searched the bibliographies of selected studies and meta-analyses to identify further eligible studies to be included in this systematic review. Only full articles in English or articles that have English translations were included.

### 2.2. Inclusion Criteria

The eligibility criteria of our systematic review and meta-analysis included the following:(1)Studies that were cohort studies (prospective or retrospective), case–control studies, or randomized controlled trials (RCTs) that reported and compared mortality rates between female and male patients undergoing TAVR with newer generation THVs.(2)Relative risk (RR), odds ratio (OR), hazard ratio (HR) with 95% confidence intervals (CI), or sufficient raw data to calculate the effect size of mortality rate differences between female and male patients undergoing TAVR with newer generation THVs.

### 2.3. Data Extraction and Quality Assessment Tool

Two investigators (AT and ST) independently performed data extraction. The data extraction process included extracting the relevant baseline characteristics and outcomes from included studies. If there was any discrepancy in extracted data, the disputes were resolved by consensus following discussion with a third investigator (JK), who would review the data independently and was blinded to the previous initial extracted data by the two investigators (AT and ST). After data extraction was completed, the included studies were assessed for quality. The quality of observational studies was assessed using the Newcastle–Ottawa Scale (NOS) [9]. The NOS utilizes a star system ranging from 0 to 9 to assess observational studies in 3 main domains. The three main domains included (1) recruitment and selection of the participants, (2) similarity and comparability between the groups, and (3) ascertainment of the outcome of interest among studies. The quality of each domain was assessed and given a score by investigators, and the summation of scores from the three domains represented the quality of the study. Higher scores constitute a higher study quality, with a score of 9 indicating the best quality available. For studies that were RCTs, the Cochrane Collaboration tool for assessing the risk of bias was used to evaluate the quality and risks of bias of each RCT in six domains. The six domains included (1) random sequence generation (selection bias), (2) concealment of allocation (selection bias), (3) blinding of participants and involved personnel (performance bias), (4) blinding of outcome assessment by involved personnel (detection bias), (5) incomplete report of outcome or data (attrition bias), and (6) other bias. Each domain was assessed by the investigators and was given a judgment as high, low, or unclear for individual elements from five domains [10].

### 2.4. Outcomes and Definitions

The primary outcomes of our study were early mortality and late mortality following TAVR with newer generation THVs. Early mortality was defined as mortality within 30 days after the index procedure, while late mortality was defined as mortality between 30 days and 1 year after the index procedure. Secondary outcomes were procedure-related major vascular complications. The primary outcomes, early and late mortality, and secondary outcomes were compared between the genders and calculated for effect size.

Newer generation THVs were described as Edwards Sapien 3, Medtronic Corevalve Evolut R, and Medtronic Evolut Pro.

### 2.5. Statistical Analysis

This meta-analysis was performed using a random-effects model. The extracted studies were excluded from the analysis if they did not justify an outcome in each cohort. Odds ratio was used to determine the outcome difference between the two groups. Q-statistic and I^2^ statistic were used to assess evidence of heterogeneity [11]. The I^2^ statistic ranges in value from 0 to 100%, with I^2^ < 25% indicating low heterogeneity, I^2^ = 25–50% indicating moderate heterogeneity, and I^2^ ≥ 50% indicating substantial heterogeneity [12]. Publication bias was assessed using a funnel plot, Begg’s test, and Egger’s test [13,14]. The *p*-value < 0.05 in publication bias tests was suggestive of significant publication bias. Sensitivity analysis was also performed by excluding one study at a time to assess the influence of each study on the overall results of the meta-analysis, as described by Patsopoulos et al. [11,15]. All analyses were conducted using Review Manager 5.1 (Nordic Cochrane Centre, The Cochrane Collaboration, Copenhagen, Denmark) and STATA software (version 14 STATA Corp, College Station, TX, USA).

## 3. Results

### 3.1. Literature Search

The initial literature search identified a total of 3689 studies from the MEDLINE (1299 studies) and Embase (2390 studies) databases. At this stage, we excluded 1314 studies due to duplication, leaving 2375 studies for abstract screening. Two thousand three hundred and seventy-four studies were further excluded as they were not cohort studies, case–control studies, or RCTs, or did not include the population of interest of our study objective. This left 66 studies for full-text review. Sixty-one studies were further excluded due to the following: (1) no outcomes of interest, (2) no comparison between gender groups, (3) the same author group with the same database, (4) studies conducted in the older generation THVs. Therefore, five studies were included in the meta-analysis [16,17,18,19,20]. The PRISMA flow diagram outlining the literature search process is shown in Figure 1. The PRIMA checklist is shown in Appendix A.

### 3.2. Description of Included Studies

A total of 47,933 patients from 5 prospective observational studies (4 studies from the United States and 1 study from Europe) from 2018 to 2022 were included in the statistical analysis. Two studies used the same database, but one reported early mortality [19], and another one reported one-year mortality [20]. The participants involved 21,073 women and 26,860 men. The mean age was 80 ± 9.1 and 78.5 ± 9.4 in women and men, respectively (*p* = 0.08). The male patients had a greater history of coronary artery disease (CAD) than the female patients. However, there were no differences in history of atrial fibrillation (AF), diabetes mellitus (DM), hypertension (HTN), stroke, or chronic kidney disease (CKD) between the two genders. Ninety-six percent received TAVR via the transfemoral approach. The summary of the baseline characteristics of the included studies is shown in Table 1.

### 3.3. Quality Assessment Tool

The NOS of the included studies is described in Appendix A. As we did not include any RCTs, we did not use the Cochrane Collaboration tool for assessing the risk of bias.

### 3.4. Meta-Analysis Results

Early mortality was reported in four studies and occurred in 1.65% and 1.01% of women and men, respectively. The female gender was associated with higher early mortality rates after TAVR with newer generation THVs (OR = 1.53, CI 1.31–1.79, *p* < 0.001, I^2^ = 0%) (Figure 2). Late mortality was reported in three studies and occurred in 10.15% and 12.25% of women and men, respectively. There was no difference in late mortality between the two genders (OR = 0.78, CI 0.61–1.00, *p* = 0281, I^2^ = 22%) (Figure 3). Vascular complications were reported in three studies and occurred in 2.93% and 2.06% of women and men, respectively. The female gender was found to have a higher rate of vascular complications after TAVR (OR = 1.43, CI 1.23–1.65, *p* < 0.001, I^2^ = 11%) (Figure 4).

#### 3.4.1. Sensitivity Analysis

We conducted a sensitivity analysis for each outcome by excluding one study at a time to assess the stability of the results of the meta-analysis. For every outcome, none of the results were significantly altered, as the results were similar to those of the main meta-analysis, indicating that our results were robust.

#### 3.4.2. Publication Bias

Although we initially intended to evaluate for publication bias, as there were fewer than 10 studies included in our analysis, we did not perform a funnel plot as it was inevitably insufficient to reject the assumption of no funnel plot asymmetry [21,22].

## 4. Discussion

This is the first systematic review and meta-analysis to date to assess the effects of gender on patients undergoing TAVR with the newer generation THVs. We found that gender disparities do persist in early outcomes similar to the early generation THVs including higher risks of vascular complications and 30-day mortality rates. However, there was no difference in one-year mortality between genders.

Severe AS, when present with symptoms (stage D AS) including angina, syncope, or heart failure, is associated with increased morbidity and mortality up to 50% within two years after symptom onset [23]. Aortic valve replacement is indicated as a class I recommendation in patients with severe AS who are experiencing symptoms [24]. In patients with severe AS without symptoms (stage C AS), guidelines suggest that if additional factors are present, including a decrease in the left ventricular ejection fraction <50%, hypotension during an exercise treadmill test, aortic valve maximum velocity >5 m/s, B-type natriuretic peptide >3 times the upper normal limit, and rapid disease progression, aortic valve replacement is also recommended [24]. Historically, surgical aortic valve replacement (SAVR) has been the mainstay of treatment for aortic valve replacement in patients with severe AS that can reduce symptoms and improve outcomes [25]. However, as severe AS is a degenerative disease, patients with symptomatic severe AS usually have an advanced age and high comorbidities with prohibitive surgical risks. Real-world evidence suggested that up to 30% of patients with severe symptomatic AS were turned down for SAVR, given the high surgical risks [26]. The revolutionary TAVR was initially introduced in 2002 and was demonstrated in the landmark PARTNER (The Placement of Aortic Transcatheter Valves) trial that in patients with severe AS who could not undergo SAVR, TAVR decreased all-cause mortality and composite outcomes of all-cause mortality, cardiac hospitalization, and symptoms, compared to medical therapy alone [23]. Subsequent head-to-head comparisons between SAVR and TAVR were demonstrated in the PARTNER 2 trial in 2016 [27] and the PARTNER 3 trial in 2019 [28,29]. The PARTNER 2 trial showed that in intermediate-risk patients with severe AS undergoing aortic valve replacement, there were no differences in the composite outcome of death or disabling stroke between TAVR and SAVR. The PARTNER 3 trial further strengthened the evidence of TAVR in low-risk patients, revealing that the TAVR group had significantly lower 30-day stroke or death and 1-year composite outcome of death, stroke, and 1-year rehospitalization than the SAVR group. In the two-year follow-up data of the PARTNER 3 trial, the differences in the composite outcome persisted at the two-year follow-up; however, the initial reduction in death and stroke in the TAVR group was no longer statistically significant and was off-set by an incidence of valve thrombosis in the TAVR group [30].

Despite the robust safety and beneficial data of TAVR, complications still occur from this procedure. Common procedural-related complications include stroke, vascular complication, conduction system injury requiring a permanent pacemaker, bleeding, and acute kidney injury [31]. Unoptimized valve sizing or placement can also lead to significant paravalvular leak (PVL) or patient–prosthetic mismatch (PPM) [32]. More devastating adverse events were rare but included device landing zone rupture, cardiac tamponade, coronary artery occlusion, and cardiac arrest.

Though complications were reported to be as high as 30% earlier in the TAVR era [31], the complications from TAVR have decreased substantially over time [7]. This is likely due to increased operators’ experiences, a more refined procedural protocol, and the development of the new generation THVs. The introduction of the new generation THVs, including Sapien 3, Corevalve Evolut R, and Evolut Pro, has advanced the TAVR technology and additionally reinforced procedural safety. A recent systematic review and meta-analysis of 14 studies comparing the older and newer generation THVs indicated that the newer generation THVs had better clinical efficacy and fewer postprocedural complications directly compared to the older generation THVs, including major vascular complications, major bleeding, significant PVL, acute kidney injury, and all-cause mortality [8]. However, there were no statistically significant differences in stroke, acute myocardial infarction, and new permanent pacemaker implantation between the older and newer generation THVs. Several reasons make the new generation THVs safer and have fewer complications. Firstly, newer generation THVs delivery systems can deliver the bioprosthetic valve through a significantly smaller sheath size as small as 14 Fr, which leads to a substantially decreased risk of vascular complications compared to the previous generation THVs, which use a bigger sheath size up to 28 Fr [33]. Secondly, the newer generation THVs, including Edwards SAPIEN-3 and Evolut Pro valves, have the development of an outer skirt that can deliver a better seal, resulting in a lower PVL risk, which is a common risk factor for long-term mortality in patients undergoing TAVR. Even in patients who develop at least moderate PVL after TAVR, the study showed that this population had a regression of PVL at the one-year follow-up [34,35]. Additionally, the pacemaker placement rates decreased with SAPIEN-3 and Evolut R, which deployed the valves at higher implantation heights. This was believed to cause less high-grade atrioventricular block requiring a permanent pacemaker [36]. There are ongoing debates and conflicting data on stroke risk comparison between the old and new generation THVs, as the lower stroke rates following the new generation THVs can be secondary to the concomitant use of cerebral protection systems. Gender significantly impacts the procedure and complication rates. In patients undergoing TAVR with older generation THVs, females, in addition to other well-known factors including older age, obesity, chronic kidney disease, peripheral vascular disease with significant calcification, and vascular tortuosity, were found to have higher risks of vascular complications [37]. We found that despite the better safety profile of the newer generation THVs, the gender disparities in the outcomes persist with the newer generation THVs, specifically vascular complications and short-term outcomes. Similar to the use of early generation THVs, the higher rates of vascular complications in female patients were thought to be related to the diameter of the sheath size used during the procedure and greater technique difficulty due to smaller vascular anatomy including aortic root size [38]. The study by Barbanti et al. showed that the insertion of a larger sheath during transfemoral TAVR was associated with a significantly higher risk of vascular complications [39]. This finding was supported by Hayashida at al. who described that the sheath-to-femoral artery ratio was a predictor of vascular complications [40]. The smaller arterial diameter in women may lead to a higher sheath-to-femoral artery ratio, which resulted in the higher rates of vascular complications observed in the published data. Although the newer generation THVs were developed to be more suitable for female bodies with smaller sheath diameters from the initial sheath sizes of 24–26 Fr to 14–16 Fr, we still found a higher rate of vascular complications among females undergoing TAVR with the newer generation THVs. This suggests that there are factors other than sheath size contributing to the risk of vascular complications from TAVR. Good pre-procedure access site planning and access site closure, as well as consideration for alternative access if a femoral approach is borderline acceptable, might be able to reduce the rates of vascular complications, especially in female patients.

Regarding early mortality, there are several possible reasons for the higher early mortality rate, but not late mortality, in female patients. Although the higher rate of perioperative complications, including bleeding and vascular complications, may be one of the explanations for higher early mortality rates, evidence suggests that mortality risks are multifactorial. Greater ventricular wall thickness, smaller left ventricular systolic cavity, and narrow outflow tract with flow acceleration in women were found to be associated with higher postoperative mortality rates in female patients following aortic valve replacement [41,42]. Moreover, a higher prevalence of pulmonary hypertension (PH) in female patients could be one of the factors for higher early mortality rates [43]. Alushi et al. reported higher risks of 30-day mortality after TAVR in patients with residual PH (hazard ratio 3.49, *p* < 0.001) [44]. Kjønås et al. similarly demonstrated that systolic pulmonary arterial pressure >60 mmHg was one of the independent predictors of 30-day mortality [45]. Regardless if patients survive the peri-operative and 30-day period, females are found to have a similar prognosis to males, if not better. Studies with a longer follow-up duration are needed to validate whether gender disparity exists at follow-up after one year.

Limitations: This study is not without limitations. Firstly, this meta-analysis was mainly driven by observational studies not adjusted for confounders. This carries a risk of confounding bias. Secondly, though several studies reported gender gaps in other adverse events following TAVR, we had insufficient data to analyze other adverse events including bleeding, stroke, and new pacemaker implantation.

## 5. Conclusions

With the safer profile of TAVR with newer generation THVs, our systematic review and meta-analysis reviewed worse early outcomes in the female gender. The female gender was associated with a higher early mortality rate and vascular complication after TAVR with newer generation THVs. There were no differences between the genders in late mortality.

## Figures and Tables

**Figure 1 medsci-11-00033-f001:**
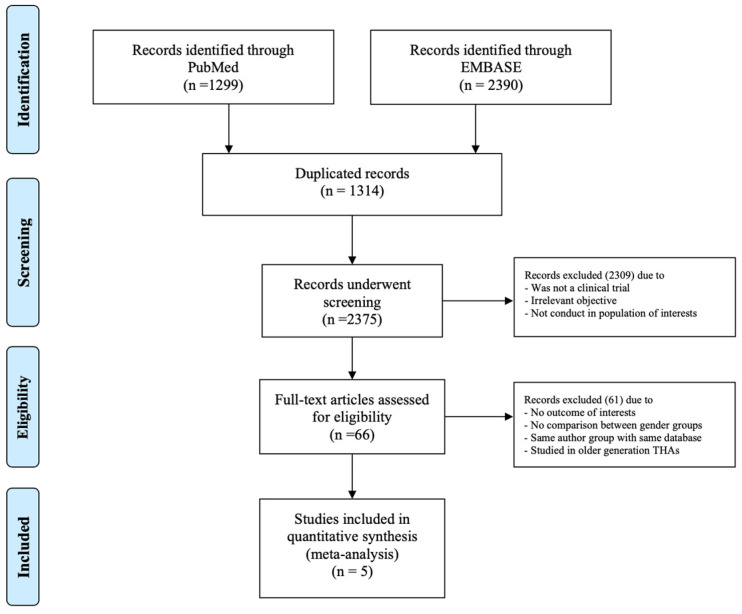
PRISMA flow diagram illustrating the study selection process.

**Figure 2 medsci-11-00033-f002:**
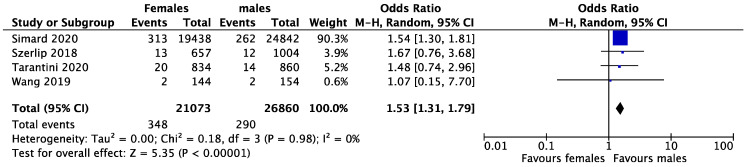
Forest plot for pooled early mortality after TAVR between females and males. 95% CI = 95% confidence interval; Ch^2^ = Chi-square; I^2^ = I-square; P = *p*-value; Tau^2^ = Tau-square; Z = Z test.

**Figure 3 medsci-11-00033-f003:**
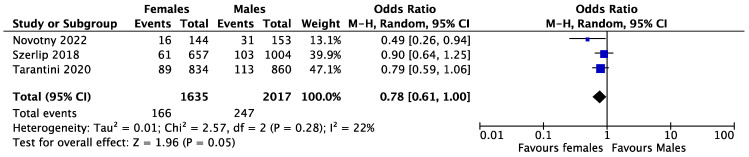
Forest plot for pooled late mortality after TAVR between females and males. 95% CI = 95% confidence interval; Ch^2^ = Chi-square; I^2^ = I-square; P = *p*-value; Tau^2^ = Tau-square; Z = Z test.

**Figure 4 medsci-11-00033-f004:**
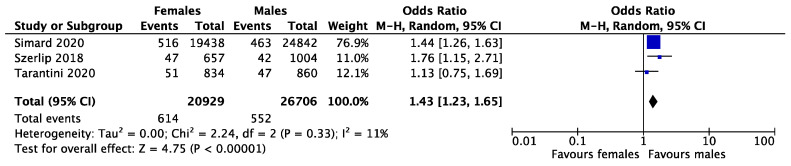
Forest plot for pooled vascular complications after TAVR between females and males. 95% CI = 95% confidence interval; Ch^2^ = Chi-square; I^2^ = I-square; P = *p*-value; Tau^2^ = Tau-square; Z = Z test.

**Table 1 medsci-11-00033-t001:** Baseline Characteristics for TAVR.

Study (First Author, Year)	Country	Study Type	Number of Patients			Age (Mean ± SD)		CAD (%)		AF (%)		DM (%)		HTN (%)		Stroke/TIA (%)		CKD (%)		Follow-Up (Year)	Access (%)		Valve (%)	
			Total	F	M	F	M	F	M	F	M	F	M	F	M	F	M	F	M		TF	Non-TF	Balloon	Self- Expanding
Novotny, 2022 *	USA	P	297	144	153	81 ± 8	79 ± 9	16.7	30.1	27.1	45.1	34.3	43.1	N/A	N/A	16.7	14.4	N/A	N/A	1	100	0	78	22
Simard, 2021	USA	P	44,280	19,438	24,842	79.8 ± 9.2	78.3 ± 9.8	61	76.9	33.9	37.4	38.9	42.2	N/A	N/A	13	13.1	32	39.6	During hospitalization	99.2	0.8	N/A	N/A
Szerlip, 2018	USA	P after PARTNER II S3 trial	1661	657	1004	82.5 ± 7.2	82 ± 7	58.9	80.5	N/A	N/A	31.2	35.2	92.7	93.1	16.4	18.4	6.5	10.9	1	86.9	13.1	100	0
Tarantini, 2020	European countries	P	1694	834	860	82.7 ± 6.29	80.8 ± 6.91	39.8	57.6	19.9	24.8	27.2	30.2	82.3	80.8	N/A	N/A	N/A	N/A	4	100	0	100	0
Wang, 2019 *	USA	P	298	144	154	81 ± 8	79 ± 9	16.7	30.5	27.1	44.8	34.3	43.5	N/A	N/A	16.7	14.3	N/A	N/A	During hospitalization	99.7	0.3	78.2	21.8
Total	N/A	N/A	47,933	21,073	26,860	80 ± 9.1	78.5 ± 9.4	44	61	23.5	34.8	32.8	37.8	87.5	87	15.4	15.2	25.3	4.8	N/A	97	3.6	98.2	1.8
*p*-value	N/A	N/A	N/A			0.08		0.01		0.06		0.5		0.8		0.5		0.3		N/A	<0.001		<0.001	

AF = atrial fibrillation; CAD = coronary artery disease; CKD = chronic kidney disease; DM = diabetes mellitus; HTN = hypertension; F = female; M = male; N/A = not applicable; P = prospective study; TF = transfemoral; TIA = transient ischemic attack. * Same database; calculation was based on Wang 2019.

## Data Availability

No new data were created.

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
