# Peer review of "Gender Disparities after Transcatheter Aortic Valve Replacement with Newer Generation Transcatheter Heart Valves: A Systematic Review and Meta-Analysis"

_medsci, 2023, doi:10.3390/medsci11020033_

Round 1
Reviewer 1 Report
-Gender differences in percutaneous procedural outcomes is a hot topic, even if differences are sometimes driven by anatomical and epidemiological characteristics: please explain that in the papers selected by the review, female patients showed smaller access arteries and that differences in major vascular access complication might be explained by smaller arteries diameter rather than gender per se.
-Differences in mid-term outcomes in elderly populations might be explained by longer life obeserved among female gender in the general population: please comment.
-Overall, the message of the paper should be that with new generation devices overall mortality and major complications decreased in comparison to previous studies (see data cited in the introduction); such a decrease was observed in a greater extent among the female population (see morality and complications reported by the metanalysis in comparison with data cited in the introduction).
-The residual difference in primary outcomes between men and women, altough still present, is thus smaller with new generation devices to such an extent that it might be considered marginal, expecially if results are not weighted for confounders.
Author Response
- Gender differences in percutaneous procedural outcomes is a hot topic, even if differences are sometimes driven by anatomical and epidemiological characteristics: please explain that in the papers selected by the review, female patients showed smaller access arteries and that differences in major vascular access complication might be explained by smaller arteries diameter rather than gender per se.
- Appreciate reviewer’s comment. We discussed the higher vascular complications in females may contribute from smaller artery diameters in females and a higher sheath-to-femoral artery ratio; however, newer generation THVs were developed with smaller sheath sizes, females still were associated with higher vascular complications, so other factors need to be explored in the discussion part as below.
“Similar to the use of old generation THVs, the higher rates of vascular complications in female patients were thought to be related to the diameter of the sheath size used during the procedure and greater technique difficulty due to smaller vascular anatomy including aortic root size [37]. The study by Barbanti et al. showed that the insertion of a larger sheath during transfemoral TAVR was associated with significantly higher vascular complications [38]. This finding was supported by Hayashida at al. who described that the sheath-to-femoral artery ratio was a predictor of vascular complications [39]. The smaller arterial diameter in women may lead to a higher sheath-to-femoral artery ratio which resulted in the higher vascular complications observed in the published data. Although the newer generation THVs were developed to be more suitable for female bodies with smaller sheath diameters from initial sheath sizes of 24-26 Fr to 14-16 Fr, we still found the higher vascular complications among females undergoing TAVR with the newer generation THVs. This suggestes that there are factors other than sheath size contributing to the risk of vaswere cular complications from TAVR.”
- Differences in mid-term outcomes in elderly populations might be explained by longer life obeserved among female gender in the general population: please comment.
- Appreciate reviewer’s comment. In our study, we found higher vascular complications and 30-d mortality in females, but there was not difference in 1-year mortality. We added the analysis comparing baseline age between 2 genders in the table 1 and found no difference in age undergoing TAVR between females and males. Longer life expectancy in females might not be an explanation in those findings.
- Overall, the message of the paper should be that with new generation devices overall mortality and major complications decreased in comparison to previous studies (see data cited in the introduction); such a decrease was observed in a greater extent among the female population (seemorality and complications reported by the metanalysis in comparison with data cited in the introduction).
- Appreciate reviewer’s comment. Objectives of this study were to assess if gender disparities persist after the development of newer generation THVs. Our study found that even though TAVR with newer generation THVs was associated with better outcomes compared to early generation THVs, our study still found that females were associated with higher early mortality and vascular complications. We added this point in the conclusion as below:
“With the safer profile of TAVR with newer generation THVs, our systematic review and meta-analysis reviewed worse early outcomes in female gender: the female gender was associated with a higher early mortality and vascular complication after TAVR with newer generation THVs. Hoewer, there were no differences between genders in late mortality.”
- The residual difference in primary outcomes between men and women, altough still present, is thus smaller with new generation devices to such an extent that it might be considered marginal, expecially if results are not weighted for confounders.
- Appreciate reviewer’s comment. We added analysis comparing baseline characteristics between 2 genders and found no significant difference expect higher prevalence of CAD in males. However, there were other factors that we were unable to compare which can be confounders. We discussed this point in the limitations as below:
“This study is not without limitations. Firstly, this meta-analysis was mainly driven by observational studies not adjusted for confounders. This carries a risk of confounding bias.”
Reviewer 2 Report
In this paper the authors have investigated gender difference in clinical outcomes after TAVI procedure using new generation transcatheter prosthesis. According to this metanalisys female gender is associated with higher 30 day mortality and vascular complication after TAVI with new generation transcatheter prosthesis, whereas there are no differences in 1-year mortality between men and women. The topic is very intersting, the study is well written and the statistical method is valid and correctly applied but there are some queries.
1) The authors have reported only data about clinical outcomes but the difference in clinical outcome may be related to associated comorbidities at the time of intervention and to woman later referral. Therefore the different risk profile between man and women in the included studies may be reported and an adjusyed survival analysis may be performed.
2) in the study new generation transcatheter prosthesis are included such as Sapien 3, Evolut R and Evolut Pro. In the discussion the authors have idenified in teh Sapien 3 outer skirt a mechanism to reduce paravalvular leak but the Evolut Pro have also a outer skirt.
3) Several study have reported superior outcomes with TAVI in women compared with men, partially due to longer life expectancy in women
Author Response
1) The authors have reported only data about clinical outcomes but the difference in clinical outcome may be related to associated comorbidities at the time of intervention and to woman later referral. Therefore the different risk profile between man and women in the included studies may be reported and an adjusyed survival analysis may be performed.
- Appreciate reviewer’s comment. We added analysis comparing baseline characteristics between 2 genders as shown in table 1. There was no difference in age, atrial fibrillation, DM, HTN, stroke, and CKD between women and men, but there were higher rates of CAD in men. Unfortunately, we were unable to perform adjusted survival analysis due to unavailable data.
2) in the study new generation transcatheter prosthesis are included such as Sapien 3, Evolut R and Evolut Pro. In the discussion the authors have idenified in teh Sapien 3 outer skirt a mechanism to reduce paravalvular leak but the Evolut Pro have also a outer skirt.
- Appreciate reviewer’s comment. We added discussion about the development of outer skirt of Evolut Pro in the discussion as below:
“Secondly, the newer generation THVs including Edwards SAPIEN -3 and Evolut Pro valves have the development of outer skirt which can deliver a better seal, resulting in a lower PVL risk, which is a common risk factors for long-term mortality in patients undergoing TAVR. Even in patients who develop at least moderate PVL after TAVR, study showed that this population had regression of PVL at 1 year follow-up (33, 34).”
3) Several study have reported superior outcomes with TAVI in women compared with men, partially due to longer life expectancy in women
- Appreciate reviewer’s comment. Some studies reported better 1-year survival rates in females after TAVR. However, those studies have included TAVR with early generation THVs in the analysis. In our study, we studied in only post-TAVR with newer generation THVs including Sapien 3, Evolut R, and Evolut Pro and found there was no difference in 1-year mortality between 2 genders. We also performed analysis comparing baseline characteristics between genders and we found that there was no difference in age between women and men.
Round 2
Reviewer 1 Report
The Authors extensevely answered the questions risen, the paper is acceptable in the present form
Author Response
We appreciate the reviewer's comment.
Reviewer 2 Report
I have read the revised version. According to my opinion the revised manuscript may be accept in present form.
Author Response
We appreciate the reviewer's comment.